# Developing an Echocardiography-Based, Automatic Deep Learning Framework for the Differentiation of Increased Left Ventricular Wall Thickness Etiologies

**DOI:** 10.3390/jimaging9020048

**Published:** 2023-02-18

**Authors:** James Li, Chieh-Ju Chao, Jiwoong Jason Jeong, Juan Maria Farina, Amith R. Seri, Timothy Barry, Hana Newman, Megan Campany, Merna Abdou, Michael O’Shea, Sean Smith, Bishoy Abraham, Seyedeh Maryam Hosseini, Yuxiang Wang, Steven Lester, Said Alsidawi, Susan Wilansky, Eric Steidley, Julie Rosenthal, Chadi Ayoub, Christopher P. Appleton, Win-Kuang Shen, Martha Grogan, Garvan C. Kane, Jae K. Oh, Bhavik N. Patel, Reza Arsanjani, Imon Banerjee

**Affiliations:** 1Mayo Clinic Arizona, Scottsdale, AZ 85054, USA; 2School of Computing and Augmented Intelligence, Arizona State University, Phoenix, AZ 85281, USA; 3Mayo Clinic Rochester, Rochester, MN 55905, USA

**Keywords:** deep learning, LV wall thickness, echocardiography

## Abstract

Aims:Increased left ventricular (LV) wall thickness is frequently encountered in transthoracic echocardiography (TTE). While accurate and early diagnosis is clinically important, given the differences in available therapeutic options and prognosis, an extensive workup is often required to establish the diagnosis. We propose the first echo-based, automated deep learning model with a fusion architecture to facilitate the evaluation and diagnosis of increased left ventricular (LV) wall thickness. Methods and Results: Patients with an established diagnosis of increased LV wall thickness (hypertrophic cardiomyopathy (HCM), cardiac amyloidosis (CA), and hypertensive heart disease (HTN)/others) between 1/2015 and 11/2019 at Mayo Clinic Arizona were identified. The cohort was divided into 80%/10%/10% for training, validation, and testing sets, respectively. Six baseline TTE views were used to optimize a pre-trained InceptionResnetV2 model. Each model output was used to train a meta-learner under a fusion architecture. Model performance was assessed by multiclass area under the receiver operating characteristic curve (AUROC). A total of 586 patients were used for the final analysis (194 HCM, 201 CA, and 191 HTN/others). The mean age was 55.0 years, and 57.8% were male. Among the individual view-dependent models, the apical 4-chamber model had the best performance (AUROC: HCM: 0.94, CA: 0.73, and HTN/other: 0.87). The final fusion model outperformed all the view-dependent models (AUROC: HCM: 0.93, CA: 0.90, and HTN/other: 0.92). Conclusion: The echo-based InceptionResnetV2 fusion model can accurately classify the main etiologies of increased LV wall thickness and can facilitate the process of diagnosis and workup.

## 1. Introduction

Increased left ventricular (LV) wall thickness is frequently encountered in transthoracic echocardiography (TTE) studies, and the common etiologies include hypertrophic cardiomyopathy (HCM), cardiac amyloidosis (CA), and other conditions such as hypertensive heart disease (HTN) [1,2]. In clinical practice, TTE is usually considered the first-line screening/diagnostic tool for increased LV wall thickness, and certain echocardiographic features have been described for the etiologies mentioned above, but some features may overlap among different etiologies. Although trained echocardiographic experts can make a preliminary diagnosis based on certain features, extensive workup including advanced cardiac imaging or even myocardial biopsy is usually required to establish a diagnosis [3,4,5]. Concerning the substantial differences in available therapeutic options and prognosis of each disease, if patients were not referred for a proper diagnostic test initially, adverse events could happen and lead to extra medical costs and a worse prognosis [3,5]. In practice, patients with CA can be easily mislabeled as HCM, and miss the opportunity to maximize the benefit of an earlier therapeutic intervention [5]. 

With the recent advance in artificial intelligence (AI) technology, image-based deep learning algorithms have been applied in the field of cardiovascular imaging [6,7]. Specifically, echocardiography-based approaches have been applied in auto-segmentation, auto-strain imaging, predicting in-hospital mortality, auto-view classification, as well as interpretation of the study [8,9,10,11,12]. An echocardiography-based deep learning model for the differential of increased LV wall thickness would greatly facilitate the evaluation process by suggesting the most high-yield diagnosis. However, the development of such a model within the research boundary remains at an early stage [13,14,15]. Zhang et al. developed an automated, multi-function framework; however, the core model used was a relatively shallow convolutional neuronal network model [15]. Recently, Duffy et al. also established an Echonet-LVH model that can differentiate HCM and CA from other etiologies of increased LV wall thickness [13]. The classification task in the above studies mainly relied on the AP4 view [13,14,15], which may have missed important features for HCM cases, such as the systolic anterior motion of the mitral apparatus and left ventricular outflow obstruction in the apical 3 chamber view [16]. 

We hypothesized that a TTE image-based deep learning algorithm with a fusion architecture could accurately classify the important etiologies (HCM, CA, and HTN/others) of increased LV wall thickness. Our group has proposed a fusion architecture that has the potential to improve the overall model performance [17,18] by incorporating 6-standard echocardiography views (apical 2–,3–,4–chamber, parasternal long axis, parasternal short axis at the mitral valve, and the mid-ventricle levels) from TTE. We developed an automated, AI-enabled framework for the classification task of increased LV wall thickness using frame-by-frame analysis, which favors a faster inference time for clinical usage. This approach also allows the training of a complex deep learning model with fewer individualized studies, as every frame is considered a separate data point. 

## 2. Methods

### 2.1. Population Selection

This retrospective study was approved by the Mayo Clinic Institutional Review Board (IRB). Patients with established diagnoses for increased LV wall thickness (hypertrophic cardiomyopathy (HCM), cardiac amyloidosis (CA), and hypertensive heart disease (HTN)/others) between 1/2015 and 11/2019 at Mayo Clinic Arizona were identified. For HCM patients, the diagnosis was established according to the guidelines, and cardiac magnetic resonance imaging was obtained [3]. For CA patients, the diagnosis was confirmed by technetium Tc^99m^ pyrophosphate scanning or endomyocardial biopsy. Cases were not further divided into the AL- and TTR- subtypes. The HTN/other category was defined as cases that underwent a cardiac magnetic resonance imaging (MRI) for the indication of increased LV wall thickness but were deemed not HCM or CA. The initial review identified 305 HCM patients, 244 CA patients, and 254 HTN/other cases. After patients were identified, six standard TTE views, including apical 2-,3-,4- chamber views (AP2, AP3, and AP4), parasternal long-axis view (PLAX), and parasternal short-axis view (at the mitral valve (PSAX_M) and mid-left ventricular (PSAX_V) levels) were collected. The cases were excluded for missingness, incompleteness, or suboptimal image quality in any of the 6 views. Of those patients, 494 qualified (164 CA, 163 HCM, and 167 HTN/other) were used for this study. Each study consisted of a combination of the six standard TTE views (AP2, AP3, AP4, PLAX, PSAX_M, PSAX_V) with an average of 342 echocardiogram video frames per study. These studies were split into train/validation/test sets with a 72/18/10 split at the study level, with each set containing 358 (119 CA, 117 HCM, and 122 HTN/other), 90 (30 CA, 30 HCM, and 30 HTN/other), 48 (16 CA, 16 HCM, and 16 HTN/other) studies, respectively. Table 1 describes the selected cohort. Frame-wise train/validation/test splits were 138,067/27,272/22,182 images, respectively. 

### 2.2. Proposed Fusion Model Architecture

Figure 1 is a diagrammatic representation of the proposed fusion framework that highlights the core processing blocks. The trained framework is designed to directly read the TTE video clips, and produce a probabilistic diagnosis at the exam level. In addition, the framework also allows the extraction of framewise predictions for model interpretability. In the following subsection, we detailed each component and emphasized its role. This study follows the Proposed Requirements for Cardiovascular Imaging Related Machine Learning Evaluation (PRIME) of JACC: Cardiovascular Imaging [19]. 

### 2.3. Preprocessing

As a pre-processing step, echocardiogram videos that are clinically stored in the Dicom format were first processed automatically by extracting frames from the video. Once all the frames of various aspect ratios and image sizes were extracted, the frames were then converted to grayscale images and thresholded at an intensity of 29 to generate contour masks. The intensity threshold was selected empirically by analyzing intensity histogram of the mean image (average image generated from a randomly selected image subset). As seen in Figure 2, the largest contour was then selected and used to remove any burned-in PHI (Protected Health Information) or device-related information that may be in the frames. Finally, each image was resized to 456 × 456 with bicubic interpolation. Such resizing without maintaining the aspect ratio may generate squeezed images for rectangular images; however, extracted masks for the images included in the study are mostly square and thus the ultimate effect of resizing is limited in our case. 

### 2.4. View Classifier

To train an end-to-end pipeline, we designed a deep learning-based view classifier model to differentiate between the six standard echo views and an ‘unknown’ class to capture irrelevant/noisy views for this study. The ground truth labels for the images are obtained from the Echo imaging experts. The echocardiogram frames were separated into 256/90/92 at the study level for train/validation/test sets, respectively, so that there was no information leakage across patients. A pre-trained ResNeXt-101 model [20] was used as the base model and was trained with a batch size of 32, a learning rate of 0.000001, cross-entropy loss, Adam optimizer, weight decay of 0.3, over 50 epochs with early stopping (wait for 10 epochs). An optimized decision threshold was calculated for each view based on the optimal operating point collected from the receiver operating characteristic curve (ROC) plots of the validation data. If the frame was within the threshold for a view, that view label was assigned to the frame without being mutually exclusive. If a frame is classified in more than one class or not assigned a class label, that frame was determined to be an ‘unknown’ and discarded from the study. Once all the frames of one echocardiogram view/clip were classified, a majority vote was applied to get the final view prediction, as seen in Figure 1. The two annotators were cardiology fellows and were blinded to each other’s labeling.

### 2.5. View-Dependent Modeling Paradigm

We trained individual parallel models to handle different views (see Figure 1). Multiple models were initially tested for their performance, ranging from shallow AlexNet to very deep Inception-ResNetv2 [20,21,22]. After the initial tests, the ImageNet pre-trained ResNeXt-101 [20] model proved to be the best performer on the validation split and was selected as the base convolution backbone model for the classification model. ResNeXt network is constructed by stacking multiple residual blocks to achieve grouped convolution which finally results in a homogeneous, multi-branch architecture that has only a few hyper-parameters to set. The primary hyperparameter is known as “cardinality”, which is the size of the set of transformations—an essential factor in addition to the dimensions of depth and width. In this study, we used ResNeXt-101, which is 101 layers deep. The hyperparameters used for the frame-wise classification method were a batch size of 32, a learning rate of 0.000001, and a weight decay of 0.3 with cross-entropy loss for 50 epochs. The models were trained on an Nvidia RTX A5000 GPU. We trained the view-dependent models on the same train split and validated each view-dependent model on the hold-out test set. To prevent data leakage, we generated the split at the study level so that no images from the same study are mixed between train and test. 

### 2.6. Fusion Model

To further improve the overall performance of the view-dependent classification model, we generated a decision-level fusion scheme (late fusion) where we can leverage predictions from multiple models to make the final decision at the frame level [18]. The decision-level fusion scheme also allows missing views to be more robust for generalization. If a view is missing, we replace the decision with -1 to encode missing data. The study-level decision is reached via the averaging where the probability for a study is calculated by averaging for each class the frame-level decision generated by the view-dependent model represented as $\frac{1}{n}\sum^{\;}P_{1},\frac{1}{n}\sum^{\;}P_{2},\frac{1}{n}\;\sum^{\;}P_{3}$, where *n* = no of frames/video and *P_i_* represents the probability calculated for i^th^ class. We trained a logistic regression model as a meta-learner to create a weighted combination of the prediction probabilities from the six view-dependent models’ averaged probability or −1 if the view is missing. We also empirically evaluated the averaging and majority voting aggregation functions for generating fusion, but the weighted combination outperformed the simple aggregation.

### 2.7. Model Performance

#### Model Interpretability: GRADCAM and the Ablation Study

We computed the GRADCAM (Gradient-weighted Class Activation Mapping) to interpret the performance of the view-dependant models [23]. We directly obtained the activation from the final convolution layers (layer before softmax classification). The 8 × 8 GRADCAM was extrapolated and resized to be mapped to the non-processed original images for better visualization. 

We performed two distinct ablation studies to understand the impact of certain data components on the trained model in a controlled experimental setting: (1) view level: the importance of each view in the final fusion model, and how the missing views can be handled by the fusion model; (2) chamber level: chamber-related information in the view-dependent AP4 model. For view importance, one view was systematically dropped (at the study level) from the input data and replaced the corresponding model prediction with −1. For chamber-related information, since the AP4 view-dependent model had the best overall performance and is the view that can visualize all four chambers, it was chosen to perform the ablation study. We segmented the heart chamber by using a simple division of the frames in a 60:40 ratio in both the horizontal and vertical axis and obfuscated each quadrant from the AP4 view to generate the partial images. Model performance was assessed as indicated above.

## 3. Results

We summarized the cohort in Table 1 and the following subsection describes the performance of the models only on the hold-out test data. 

### 3.1. Quantitative Performance

The view classifier reached an AUROC >= 0.98 in the classification of all six views (Appendix A). The Fleiss’ Kappa inter-annotator agreement between the view classifier and three radiologists was 0.7837, and the Cohen Kappa between each pair is given in Appendix A). We evaluated the individual quantitative performance of the view-dependent models as well as the final fusion model in terms of the area under the receiver-operating characteristic curve (AUROC) (Figure 3). After selecting the optimal operating point, we calculated the precision, recall, and f1-score in Table 2. We observed the highest class-wise performance on the AP4 view (CA—0.73, HCM—0.94, and HTN/other—0.87). Among the parasternal views, PLAX outperformed (CA—0.81 HCM—0.88 and HTN/other—0.82) the others. The class-wise AUROC for the final fusion model outperformed all the individual views (CA—0.90, HCM—0.93, and HTN/other—0.92), and achieved a 93.75% true positive rate for HCM and a 75% true positive rate for both CA and HTN/other. The fusion model also achieved a high F1 score for all the classes and outperformed all the individual views, which shows that the model is able to achieve a good balance between sensitivity and specificity at the standard threshold (Table 2).

### 3.2. Model Interpretability: GRADCAM 

Figure 4 demonstrated the GRADCAM heatmap of areas with greater importance for the model decision (red- more important, green- less important) on each echo view of a representative case.

### 3.3. Model Interpretability: Ablation Study

The view-level ablation study (Figure 5) shows that the model also obtained an average 76.3% [68–93%] true positive rate if a single view is missing and is able to gather the information correlating other existing views. Missing apical window (AP) views lead to a greater drop in the model performance than short-axis views, and the primary difference is observed in CA and HTN/other categories. 

In chamber-level ablation, obfuscating quadrant 1 (LV), which represents the left ventricle (LV) of the heart has the most significant drop in the AP4 model performance (CA—0.57, HCM—0.78, and HTN/other—0.65), which reflects that LV was involved most in all these diseases. Interestingly, removing the portion of any single chamber from the input AP4 image led to a significant drop in performance in identifying CA cases (Figure 6) which reflected the nature of CA as a systemic disease involving all four chambers of the heart.

### 3.4. Weights of the Individual Model Outputs Learnt by the Meta Learner

Figure 7 summarized the weight of individual model of the meta learner and overall highest importance was achieved by the AP4 model. AP2 and short-axis view was deemed important for amyloidosis. 

## 4. Discussion

We successfully established an automatic end-to-end deep learning model framework that accurately differentiates the major etiologies of increased LV wall thickness, including HCM and CA, from the background of HTN/other diagnoses. To the best of our knowledge, this is the first deep learning study that applied multiple standard echocardiographic views with a fusion architecture and achieved an overall superior performance. Additionally, a detailed ablation study highlighting each chamber’s role along with the GradCAM analysis allows for a better interpretation of the model.

The major contributions of this work include (1) demonstrating a superior echo-based fusion model performance, especially for CA patients, without any segmentation of the images, (2) providing insights into echo view selection for future study design if not all the views are planned to be used, and (3) showing the possibility of achieving overall superior performance with significantly fewer training cases by a frame-based approach, which will facilitate the development of future models for rare conditions. 

### 4.1. An Echo-Based, End-to-End Deep Learning Model with A Fusion Architecture

While the application of deep learning is rapidly growing in medical literature, there are relatively fewer studies that have incorporated a fusion architecture [18]. A fusion architecture has the potential to improve the overall performance of deep learning models by associating multiple input channels [17,18]. This kind of approach is particularly feasible when being applied to TTE studies, as the images were obtained according to standardized views suggested by the guidelines [24,25]. Duffy et al. reported a model using the AP4 views as the input and reached an AUC of 0.83 for CA, and an AUC of 0.98 for HCM [13], while our proposed fusion model successfully pushed the AUC of CA to > 0.90 AUROC.

Our established automatic deep-learning framework allows the input of the entire echo study. After each view is identified by the view classifier, the probability output generated by each individual view-dependent model is aggregated by the meta-learner to produce the final fusion results. Compared to a video-based model [13], a frame-based model is more computationally efficient, can expand the size of the training dataset, and tolerates noisy frames in the classification process as they are aggregated by majority voting. 

With our approach, the human labeling component was minimized to the annotation of echocardiographic views and the target diagnosis. We adopted the current approach specifically to avoid segmentation of the echo images. Human annotation is known to be time-consuming and requires a high level of expertise. However, the consistency of segmentation can still vary between annotators and can hamper the training data quality [26]. Additionally, using segmented data is also subject to the loss of information from the original input. It is also reported that the model performance could depend on the quality of segmented images and may not be better than using the original images [14].

Our work also demonstrated the possibility of reaching a similar performance with significantly fewer training cases compared to the earlier work of Duffy et al. [13]. This approach will allow specific models to be developed for the diagnosis of rare conditions, for which thousands of training cases are difficult to obtain. Additionally, our model structure has the flexibility to integrate other data resources, such as electronic health records and electrocardiography, to further boost the performance.

### 4.2. Fusion: Using Information from all the Views

Zhang et al. [15] only used the PLAX and AP4 views, in which a few random frames were selected from each video. While this approach has the advantage of efficiency for computation, the information contained at a certain phase of the cardiac cycle (e.g., systolic anterior motion of mitral apparatus) can be missed in the random sampling process. Furthermore, separated models were built in their work to differentiate HCM and CA patients from their own matched control cohort; how the model(s) can differentiate cases of HCM and CA from a mixed cohort was not reported. Duffy et al. established an Echonet-LVH model that can differentiate HCM and CA from other etiologies of LVH using only PLAX and AP4 views [13]. However, their models depend on obtaining the correct segmentation of the interventricular septum, left ventricular internal dimension, and left ventricular posterior wall. An error may propagate between segmentation and final classification. Further, their model primarily extracts spatio-temporal information from echo video which can be relied on the sampling rate of the frame and thus device-dependent. 

In our study, the performance of the individual view models provided some additional insight into echo-view selection. While the PLAX view was used in all prior studies^13–15^, we observed that the performance of the PLAX model was inferior to the AP3 model (Figure 3 and Table 2). We believe this is associated with losing information from the apical area in the PLAX view. From a conventional echocardiography interpretation perspective, it is also clinically meaningful to use multiple echocardiographic views as the input data. For example, the PLAX and AP3 views contain important features for the diagnosis of HCM, such as LV outflow tract obstruction and systolic anterior motion of the mitral leaflet [3]. 

### 4.3. Clinical Application of the Model

Differentiating the etiologies and establishing a diagnosis for increased LV wall thickness remains a challenging task for clinicians [3,4,5]. CA is known to be underdiagnosed and delayed diagnosis is associated with adverse outcomes^5^. While LV strain has been proposed as a screening tool for CA [27], conducting strain analysis can be time-consuming and highly depends on imaging quality. The results can also vary between different software packages [28,29]. Furthermore, in practice, strain analysis is not routinely performed unless the reading physician has concerns about this condition. An AI-enabled, auto-populated reminder in the echo reading system can effectively address this limitation and facilitate the early diagnosis of CA. 

We foresee this model being used in daily echocardiography lab practice to improve the initial triage of increased LV wall thickness. This model will be especially useful for labs at institutions with limited diagnostic and or therapeutic resources for the above conditions. Specifically, with the high accuracy in identifying CA cases, this model will largely facilitate the early recognition of these patients and potentially improve the prognosis [5]. 

## 5. Limitations

The core limitation of this study is that it is a single-center, retrospective study with limited cases. The patients were identified at a tertiary referral center so may not be representative of the general population. Similarly, the quality of echocardiographic images could be different from images obtained in other practices or institutions. While an external validation dataset was sought, the only open-source EchoNet dataset does not contain all the view views for external validation. Given that the existing models only used a single or two views, direct comparison between the studies was not feasible. The workflow of our model did not include the automated process of detecting increased LV wall thickness conditions. However, selecting the cases with known LVH increased the model’s specificity. Compared to a video-based approach, our frame-based approach may have lost certain information about spatio-temporal relationships, but our model had reached an overall superior performance. An online calculator of this model is currently underwork, and we plan to release models and trained weights with the MIT open-source license to benchmark the performance.

## Figures and Tables

**Figure 1 jimaging-09-00048-f001:**
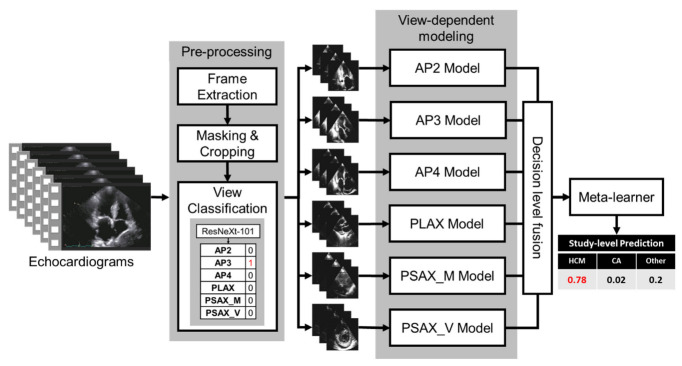
The architecture of the fusion model. Blocks represent the processing modules. AP2: Apical 2 Chamber, AP3: Apical 3 Chamber, AP4: Apical 4 Chamber, PLEX: parasternal long axis, PSAX_M: parasternal short-axis view, at the mid-ventricular level, PSAX_V: parasternal short-axis view, at the valve level levels.

**Figure 2 jimaging-09-00048-f002:**
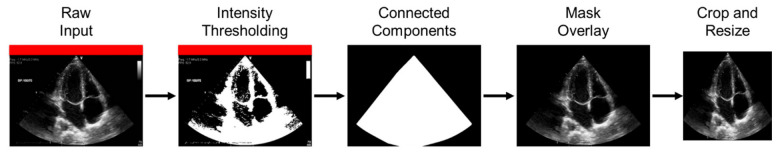
Preprocessing echocardiogram frames. Each image shows the output of the processing, and text above shows the process names. The red bars were used to anonymize the patient information for the figure.

**Figure 3 jimaging-09-00048-f003:**
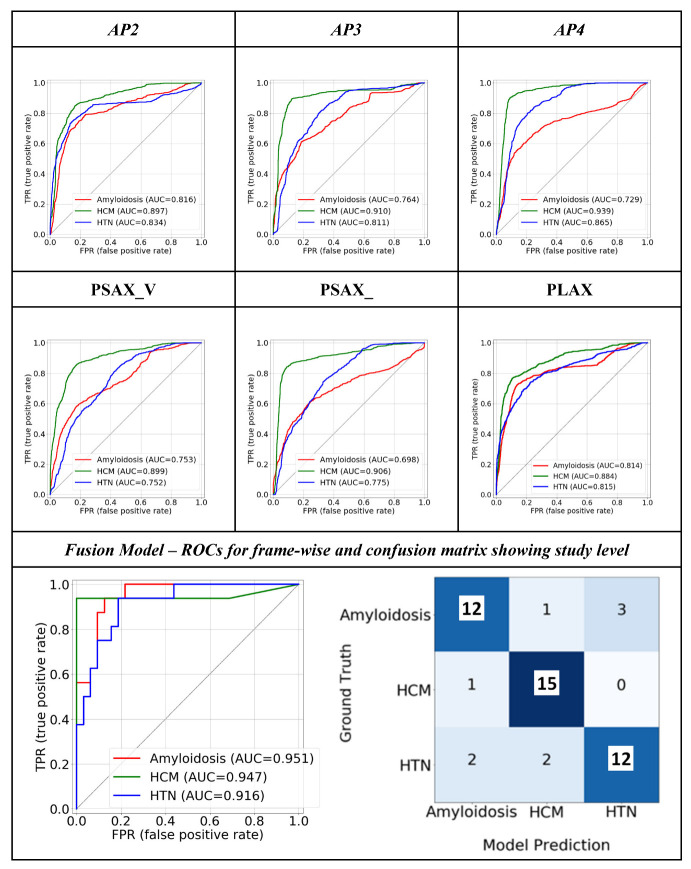
Class-wise ROC plots show the quantitative performance of the view-dependent models and final fusion model. The confusion matrix of the fusion model is also displayed. ‘Others’ category is referred to as ‘HTN.’.

**Figure 4 jimaging-09-00048-f004:**
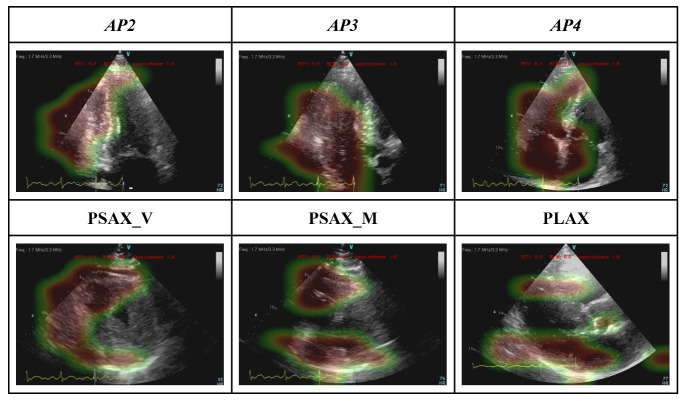
GRADCAM represents the interpretability of the view-dependent model.

**Figure 5 jimaging-09-00048-f005:**
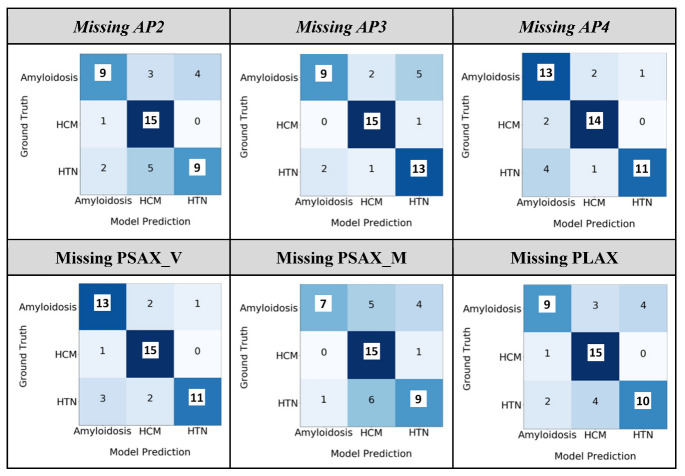
Ablation study of the fusion model at the study level by dropping one view. The ‘HTN/other’ category is referred to as ‘HTN’.

**Figure 6 jimaging-09-00048-f006:**
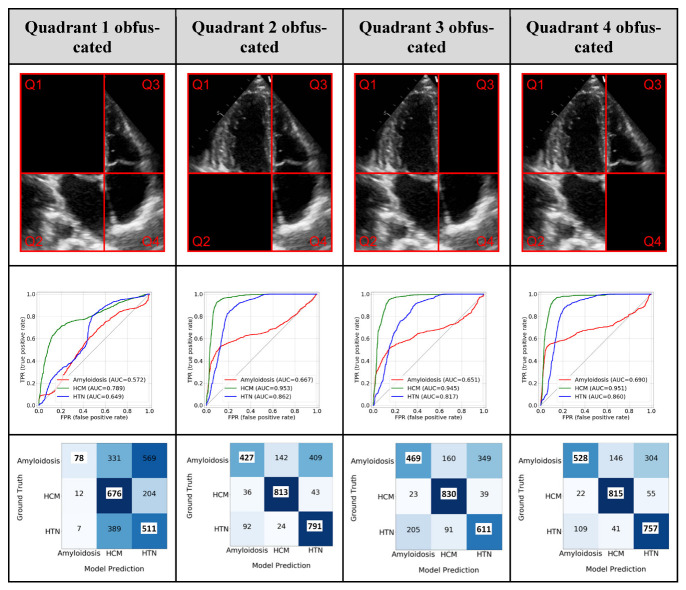
Ablation study—Removing image information. The top row shows the different quadrants (**Q1**, **Q2**, **Q3**, and **Q4**) representing different parts of the image information that was removed. The second row shows the class-wise ROC plot and the third row shows the confusion matrix using partial images. ‘Others’ category is referred to as ‘HTN’.

**Figure 7 jimaging-09-00048-f007:**
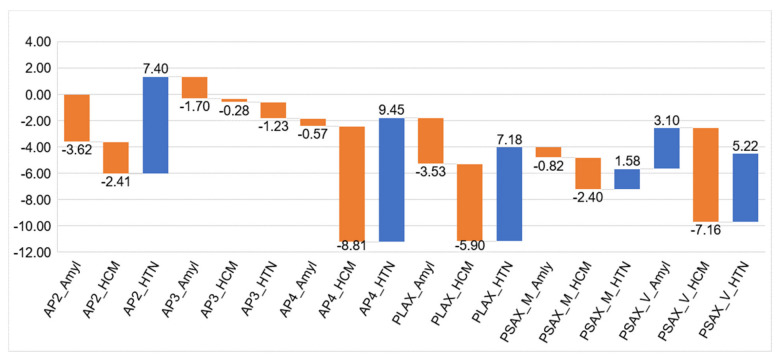
The coefficients of the meta learner for the final decision. Each model has three output channel which correspond to the three classes. Apical 2-chamber (AP2), 4-chamber (AP4), and parasternal long-axis (PLAX) views had more contributions to the final decision of the HTN/other class. Apical 4-chamber view (AP4), parasternal short- (PSAX, mid LV level) and long-axis views were more informative for the class of HCM. For cardiac amyloidosis, apical 2 chamber (AP2), parasternal long axis and parasternal short axis (mid LV level) were more informative.

**Table 1 jimaging-09-00048-t001:** Patient characteristics—age, gender, race, ethnicity, and common comorbidities.

Characteristics	Subtypes	HCM (305)	Amyloidosis (244)	HTN/Others (254)
Age		58.44 (+/−15.05)	69.25 (+/−11.15)	64.25 (+/−14.31)
Gender	Male	175 (57.37%)	196 (80.33%)	178 (70.08%)
	Female	130 (42.62%)	48 (19.67%)	76 (29.92%)
Race	White	256 (83.93%)	214 (87.7%)	218 (85.83%)
	Black or African American	17 (5.57%)	12 (4.92%)	16 (6.3%)
	American Indian	4 (1.31%)	2 (0.81%)	2 (0.79%)
	Asian	9 (2.95%)	2 (0.81%)	9 (3.54%)
	Other/Unknown	19 (6.23%)	14 (5.73%)	9 (3.54%)
Ethnicity	Hispanic or Latino	10 (3.28%)	12 (4.91%)	17 (6.69%)
	Not Hispanic or Latino	276 (90.49%)	224 (91.8%)	227 (89.37%)
	Unknown	19 (6.23%)	8 (3.27%)	10 (3.94%)
Comorbidities at the time of TTE	Hypertension	153 (38.73%)	82 (33.6%)	116 (45.67%)
	Coronary Artery Disease	71 (23.27%)	70 (28.68%)	71 (27.95%)
	Diabetics (Type I and Type II)	37 (9.37%)	21 (8.6%)	41 (16.14%)
	Chronic Kidney Disease	45 (11.4%)	51 (20.9%)	35 (13.78%)
	Congestive Heart failure	4 (1.01%)	2 (0.81%)	3 (1.18%)

**Table 2 jimaging-09-00048-t002:** Quantitate class-wise performance analysis for the single view and fusion model on the same hold out test set. 95% confidence interval was calculated using auto-bootstrapping.

Single View Models									
	**AP2**			**AP3**			**AP4**		
	Precision	Recall	F1-Score	Precision	Recall	F1-Score	Precision	Recall	F1-Score
CA	0.68 [±0.0067]	0.72 [±0.0047]	0.70 [±0.0046]	0.64 [±0.0054]	0.49 [±0.0045]	0.56 [±0.0041]	0.63 [±0.0076]	0.58 [±0.0083]	0.61 [±0.0067]
HCM	0.75 [±0.0048]	0.81 [±0.0044]	0.78 [±0.0036]	0.82 [± 0.0045]	0.85 [±0.0046]	0.83 [±0.0036]	0.77 [±0.0067]	0.93 [±0.0047]	0.84 [±0.0056]
HTN/others	0.76 [±0.0015]	0.64 [±0.0016]	0.70 [±0.0013]	0.59 [± 0.0017]	0.71 [±0.0014]	0.65 [±0.0015]	0.73 [±0.0016]	0.64 [±0.0017]	0.68 [±0.0013]
	**PLAX**			**PSAX_V**			**PSAX_M**		
	Precision	Recall	F1-Score	Precision	Recall	F1-Score	Precision	Recall	F1-Score
CA	0.70 [±0.0069]	0.70 [±0.0051]	0.72 [±0.0034]	0.57 [±0.0069]	0.62 [±0.0053]	0.59 [±0.0054]	0.57 [±0.0057]	0.58 [±0.0050]	0.57 [±0.0042]
HCM	0.85 [±0.0044]	0.66 [±0.0041]	0.74 [±0.0033]	0.76 [±0.0046]	0.78 [±0.0042]	0.77 [±0.0036]	0.87 [±0.0039]	0.78 [±0.0049]	0.82 [±0.0029]
HTN/others	0.62 [±0.0017]	0.73 [±0.0017]	0.67 [±0.0014]	0.58 [±0.0019]	0.52 [±0.0018]	0.55 [±0.0016]	0.56 [±0.0018]	0.60 [±0.0017]	0.58 [±0.0015]
**Fusion Model**									
	Precision			Recall			F1-Score		
CA	0.80 [±0.0167]			0.75 [±0.0165]			0.77 [±0.0134]		
HCM	0.83 [±0.0143]			0.94 [±0.009]			0.88 [±0.0100]		
HTN/others	0.80 [±0.0151]			0.75 [±0.01473]			0.77 [±0.01394]		

## Data Availability

The deidentified data are available upon request from Arsanjani or Banerjee with proper data usage agreement.

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
