# Peer review of "Developing an Echocardiography-Based, Automatic Deep Learning Framework for the Differentiation of Increased Left Ventricular Wall Thickness Etiologies"

_2313-433X, 2023, doi:10.3390/jimaging9020048_

Round 1
Reviewer 1 Report
The authors have established an automatic end-to-end deep learning model framework that accurately differentiates the major etiologies of increased LV wall thickness. The model well differentiated HCM and CA, and HTN/other diagnoses. Furthermore, it is interesting and valuable for assessing interpretability of the model that a detailed ablation study highlighting each chamber's role along with the GradCAM analysis.
The overall of manuscript is well designed and well written. However, the reviewer would recommend some further considerations to strength the value of this research. I hope my comment would help the readers to understand this valuable research.
Population selection: About the patients with cardiac amyloidosis (CA), it was described that “Cases were not further divided into the AL- and TTR- subtypes.”. However, in the Table 1, the patients with CA were indicated as LA. Please clarify it.
Preprocessing (Figure 2):
#1: In the processing of the crop and resize step, The aspect ratio of the image changes with respect to the original image. In the case of rectangular images, padding is generally used to resize them to squares so as not to lose the aspect ratio of the original image. Please mention the impact of this processing.
GradCAM representation: On GradCAM of AP2 in Figure 4, I can see that the models focused on right ventricular region. I wonder why the left ventricular region was not focused on. Please clarify it if you have any evidence or opinion.
Author Response
Reviewer 1:
The authors have established an automatic end-to-end deep learning model framework that accurately differentiates the major etiologies of increased LV wall thickness. The model well differentiated HCM and CA, and HTN/other diagnoses. Furthermore, it is interesting and valuable for assessing interpretability of the model that a detailed ablation study highlighting each chamber's role along with the GradCAM analysis.
The overall of manuscript is well designed and well written. However, the reviewer would recommend some further considerations to strength the value of this research. I hope my comment would help the readers to understand this valuable research.
We are glad that reviewer found our manuscript well designed and well written.
Population selection: About the patients with cardiac amyloidosis (CA), it was described that “Cases were not further divided into the AL- and TTR- subtypes.”. However, in the Table 1, the patients with CA were indicated as LA. Please clarify it.
We thank the reviewer for pointing that out and are extremely sorry for the confusion. We have updated the Table 1 header to be consistent with the rest of the paper.
Preprocessing (Figure 2):
#1: In the processing of the crop and resize step, The aspect ratio of the image changes with respect to the original image. In the case of rectangular images, padding is generally used to resize them to squares so as not to lose the aspect ratio of the original image. Please mention the impact of this processing.
The reviewer is correct. After the mask overlay, when we resize the images, we don’t particularly maintain the aspect ratio and convert the images into square (equal height and width) to increase the informative pixel in the image and reduce the empty region. This may result squeezed images; however, extracted masks for the images included in the study are mostly square and thus the ultimate effect of resizing is limited in our case.
We also added the explanation within the revised manuscript ‘Preprocessing’ section.
GradCAM representation: On GradCAM of AP2 in Figure 4, I can see that the models focused on right ventricular region. I wonder why the left ventricular region was not focused on. Please clarify it if you have any evidence or opinion.
The image is actually a revised apical 2-chamber view that contains some right ventricle tissue in the image. The model seems to focus on the ventricle septum for its decision- which is consistent with the behavior in other different views.
Importantly, the GradCAM activation may not be always perfect for precise localization as may "spill out" from the focused region since it was resized from 8x8 to fit the original images.
Reviewer 2 Report
Dear Authors,
Good paper.
This paper presents an original automated deep learning model with a fusion architecture applied to echocardiogram frames, for evaluating increased of left ventricular wall thickness in transthoracic echocardiography studies. In the considered context of this study, the results are relevant and highlight honestly the limitation of the chosen strategy.
Sincerely yours.
Author Response
Reviewer 2:
Dear Authors,
Good paper.
This paper presents an original automated deep learning model with a fusion architecture applied to echocardiogram frames, for evaluating increased of left ventricular wall thickness in transthoracic echocardiography studies. In the considered context of this study, the results are relevant and highlight honestly the limitation of the chosen strategy.
Sincerely yours.
We thank the reviewer for the encouraging comments.
Reviewer 3 Report
The authors proposed a fusion architecture that uses information from all six views by aggregating predictions from view dependent models. A view classification model was also trained to automatically label views so that an automatic, end-to-end differentiation diagnosis workflow could be achieved. The fusion model significantly outperforms all the individual view models and allows missing views for better generalization. The authors also designed ablation studies to show how the 6 views may contribute to the fusion model performance differently. The results from their ablation studies are interesting and the methodologies that authors used to evaluate “view feature importance” is inspiring. I think this is a well-designed study. I enjoyed reading the manuscript and highly recommend for publication. Below are a few minor suggestions/comments:
Page 1, line 30. Change “AUROC: CA: 0.90, HCM: 0.93, and HTN/other: 0.92” to “AUROC: HCM: 0.93, CA: 0.90, and HTN/other: 0.92” so that the order of diagnosis classes is consistent with the one in the previous sentence (page 1, line 29).
Page 1, line 114. It would be great if the authors could add more details about how the intensity threshold was determined.
Page 4, line 131. It would be great if the authors could explain more about how the decision threshold was calculated. Was it calculated based on the probabilities from the view classification model? How?
Page 4, line 134. How were the “unknown” views being used? Were they discarded?
Page 6, line 192. Shouldn’t it be AUROC >= 0.98 instead of 0.9?
Page 8, line 222. The table 2 title could be more informative.
Author Response
Reviewer 3:
The authors proposed a fusion architecture that uses information from all six views by aggregating predictions from view dependent models. A view classification model was also trained to automatically label views so that an automatic, end-to-end differentiation diagnosis workflow could be achieved. The fusion model significantly outperforms all the individual view models and allows missing views for better generalization. The authors also designed ablation studies to show how the 6 views may contribute to the fusion model performance differently. The results from their ablation studies are interesting and the methodologies that authors used to evaluate “view feature importance” is inspiring. I think this is a well-designed study. I enjoyed reading the manuscript and highly recommend for publication. Below are a few minor suggestions/comments:
Page 1, line 30. Change “AUROC: CA: 0.90, HCM: 0.93, and HTN/other: 0.92” to “AUROC: HCM: 0.93, CA: 0.90, and HTN/other: 0.92” so that the order of diagnosis classes is consistent with the one in the previous sentence (page 1, line 29).
We thank the reviewer for the suggestion. We have updated the revised manuscript.
Page 1, line 114. It would be great if the authors could add more details about how the intensity threshold was determined.
We have added more details about the intensity threshold in the revised paper.
Page 4, line 131. It would be great if the authors could explain more about how the decision threshold was calculated. Was it calculated based on the probabilities from the view classification model? How?
We collected the decision threshold based on the optimal operating point on the AUROC plots of the validation data which provides a trade-off between sensitivity and specificity.
Page 4, line 134. How were the “unknown” views being used? Were they discarded?
Yes, the unknown frames were discarded. We have added that in the revised manuscript.
Page 6, line 192. Shouldn’t it be AUROC >= 0.98 instead of 0.9?
Yes, we have updated the revised manuscript.
Page 8, line 222. The table 2 title could be more informative.
The caption of the table was redacted due to the formatting issue. We have corrected it in the revised manuscript.